# The Prevalence of Use of Various Post-Exercise Recovery Methods after Training among Elite Endurance Athletes

**DOI:** 10.3390/ijerph182111698

**Published:** 2021-11-07

**Authors:** Eduard Bezuglov, Artemii Lazarev, Vladimir Khaitin, Sergey Chegin, Aleksandra Tikhonova, Oleg Talibov, Dagmara Gerasimuk, Zbigniew Waśkiewicz

**Affiliations:** 1Department of Sports Medicine and Medical Rehabilitation, Sechenov First Moscow State Medical University (Sechenov University), 119435 Moscow, Russia; e.n.bezuglov@gmail.com; 2High Performance Sport Laboratory, Moscow Witte University, 115432 Moscow, Russia; lazarevartemii1@gmail.com (A.L.); sandratikho@gmail.com (A.T.); 3Sirius University of Science and Technology, 354349 Sochi, Russia; 4Federal Research and Clinical Center of Sports Medicine and Rehabilitation of Federal Medical Biological Agency, 121059 Moscow, Russia; 5Department of Sports Medicine and Medical Rehabilitation, Pavlov First State Medical University, 197022 Saint-Petersburg, Russia; khaitinvladimir@gmail.com; 6Football Club Zenit, 197341 Saint-Petersburg, Russia; 7Olympic Reserve Sport School, 430032 Saransk, Russia; info@smartrecovery.su; 8Department of Internal Medicine, Clinical Pharmacology and Emergency Medicine, Moscow State University of Medicine and Dentistry, 127006 Moscow, Russia; oleg.talibov@gmail.com; 9Institute of Sport Science, Jerzy Kukuczka Academy of Physical Education, 40-065 Katowice, Poland; d.gerasimuk@awf.katowice.pl

**Keywords:** massage, sauna, recovery, elite endurance athletes

## Abstract

There is now compelling evidence of the effectiveness of a range of post-exercise recovery techniques, including extended nights of sleep, cold water immersion, massage, and compression garments. Currently, limited information is available on post-exercise recovery methods used by elite endurance athletes. Therefore, this study investigated the actual methods of recovery used in this group of athletes. Google Forms were used to collect information on the recovery methods used by elite endurance track and field athletes (*n* = 153, 61.4% men, 38.6% women; average age: 22.7 ± 4.6 years). The most used methods of recovery were sauna bathing (96.7%), massage (86.9%), daytime nap (81.0%), and long night sleep (at least 9h) (61.4%). Recovery methods with proven effectiveness such as cold water immersion and compression garments were rarely used (15.0% and 7.8%, respectively). Overall, recovery methods were used more often when the tiers of the track and field athletes were higher. Massage and sauna bathing were the most used methods of post-exercise recovery among Russian endurance track and field athletes. In most cases, they were used in conjunction with short daytime nap and long night sleep. Higher tier athletes were more likely to use sauna bathing, massage, long night sleep, and daytime nap but not cold water immersion and compression garments as recovery methods; however, all these methods except for cold water immersion were widely used among elite-tier athletes.

## 1. Introduction

Endurance athletes typically alternate periods of intense training with periods of rest and recovery to achieve peak performance. Inadequate recovery can disrupt the growth and regeneration of musculoskeletal tissues and lead to overuse injury [1]. During major tournaments, the incidence of trauma and illnesses can reach up to 235 per 1000 registered track and field athletes [2]. There is strong evidence supporting the association between the use of different post-exercise recovery methods and improvements in several markers of physical performance. Sleep optimization, massage, cold immersion, compression garments, and foam rollers have been shown to be effective for the recovery of athletes of various sports [3,4,5,6,7,8,9]. Effective recovery reduces injury rates and can improve performance; however, the development and implementation of effective methods of recovery is a challenge for experts working with track and field athletes.

One of the most common consequences of excessive training load is the development of delayed muscle soreness (DOMS), accompanied by a temporary decrease in muscle strength. For many athletes of different levels, it is probably the minimization of the severity and duration of DOMS that is the main subjective indicator of recovery. Athletes with DOMS who have to continue exercising daily are advised to reduce the intensity and duration of the exercises that cause discomfort while receiving treatment [10].

Many coaches believe that massage can provide several advantages, such as increased blood flow and reduced muscle and nervous strain. These methods can increase range of motion, elasticity, general relaxation, and reduce intensity of DOMS and anxiety. All these benefits provided by massage are believed to improve athletes’ performance and reduce their risk of injury [3,4]. However, it is obvious that in addition to modifying the load in such situations, elite athletes also use other methods to minimize DOMS and recover as quickly as possible. However, here is little convincing evidence in support of the effectiveness of massage and sauna bathing despite their popularity.

In meta-analyses by Dupuy et al. [11] and Hohenauer et al. [12], data were obtained confirming the effectiveness of active recovery, massage, compression underwear, immersion in cold water, and contrast baths on the severity and duration of DOMS to varying degrees of severity. There are many more techniques used by athletes that can potentially accelerate post-exercise recovery.

Effectiveness of some methods is confirmed by data from studies of a high methodological level (meta-analyses and randomized controlled trials). However, there are methods with lack of scientific evidence of their effectiveness or contradictory data. The first group includes kinesiotaping, contrast baths, and acupuncture [13,14,15,16,17,18]. The second group of methods most likely includes cyclic compression therapy, stretching, general cryotherapy, and sauna use [19,20,21,22,23,24]. At the same time, there is little data on the actual prevalence of various recovery methods among elite athletes. It can only be assumed that they should most actively use post-exercise recovery methods with the most proven effectiveness.

Based on our observations, sauna bathing and massage have been mandatory requirements for training camps organized by coaches of Russian track and field athletes for many decades. It can be assumed that these methods are often used by this group of athletes. Considering the significant financial costs and technical difficulties necessary for its integration into the training process, a logical question arises about its appropriateness for use by professional athletes. In this study, we evaluated top-tier track and field endurance athletes, including the winners and medalists of national championships, world championships, and the Olympic Games. This cohort is among the most difficult for international researchers to gain access for research. The aim of this study was to analyze the use of various recovery methods, including massage and sauna bathing, in a group of elite endurance athletes.

We hypothesized that Russian endurance athletes incorporate sauna bathing and massage into their training regime often but follow protocols different from those previously described in published studies. We believe that such a study could help researchers to plan for future studies on the effect of sauna bathing and massage on post-exercise recovery based on the actual protocols used by young men and women athletes.

## 2. Materials and Methods

### 2.1. Participants

A total of 153 track and field Russian athletes who had performed or continue to perform at the top tier submitted their responses. All of them took part in national, continental, world championships, and the Olympic Games. Of the participants, 94 (61.4%) of them were men, and 59 (38.6%) of them were women (Table 1). The questionnaire was submitted by athletes competing in racewalking and middle- and long-distance running (distances of 800–5000 m and 10,000 m or longer, respectively; Table 2). Overall, 80 athletes were national-level athletes, i.e., participants of national track and field championships (group “national level”), and 61 athletes had participated in major international tournaments, besides national championships (Olympic Games, World and European Championships) (group “international level”). Another 12 participants had received awards in major international competitions (group “extra tier”) (Table 3).

### 2.2. Inclusion and Exclusion Criteria

Inclusion criteria:✓ written consent to participate in the study,✓ age 18 years and older,✓ participation in the national athletics teams of Russia,✓ no previous disqualifications for anti-doping rule violation or disqualification.

Exclusion criteria:✓ currently taking medication,✓ psychosocial or emotional conditions,✓ any cardiovascular or ventilatory diseases during the last 2 years.

### 2.3. Questionnaire

In a pre-test, the questions and answer choices were presented to a group of 10 semi-professional track and field athletes. Their answers were not considered in the analysis. To analyze the recovery methods used by highly competitive endurance athletes while preparing for important sporting events, the athletes were invited to complete an online questionnaire (Table 1) on their recovery methods, which was created using Google Forms. The questionnaire was made by authors and pretested on small cohort of amateur athletes. In relation to the two methods used (massage and sauna), additional questions were asked for the athletes who used them, the answers to which allowed us to determine the real protocols of their use in the studied group of athletes (Table 4 and Table 5). Responses to the questionnaire were received between October 2020 and January 2021. We assessed the frequencies of use of various methods of recovery and their association with sex, age, sporting discipline, and tier. The most common combinations of different methods of recovery were also examined.

### 2.4. Statistical Analysis

The data were processed in Microsoft Excel and analyzed using GraphPad Prism 9(GraphPad Software, San Diego, CA, USA). Kolmogorov–Smirnov test was used to assess the normality of the distribution. To compare the frequencies of use of different recovery methods, chi-squared test and Fisher’s F-test were used, and the odds ratio (OR) and 95% confidence interval (CI) were calculated. To analyze the relationship between pairs of numerical values, Mann–Whitney U-test (or Student’s t-test for normally distributed data) was used.

## 3. Results

The most used methods of recovery were sauna bathing (96.7%), massage (86.9%), daytime nap (81.0%), and long night sleep (at least 9 h) (61.4%). Interestingly, the frequencies of use of cold water immersion and compression garments were rather low (15.0% and 7.8%, respectively) despite their proven effectiveness. The frequencies of use of these methods of recovery did not differ between men and women athletes (Table 6, Figure 1, Figure 2 and Figure 3).

### 3.1. Massage

The frequency of use of massage was significantly related to the tier of the athletes (*p* = 0.024). A pairwise comparison revealed that the frequency of massage use was significantly different between the national and international tier groups, which was more common among international athletes (*p* = 0.028, OR = 3.6, 95% CI = 1.1–11.3). This method of recovery was used by all (100%) elite-tier athletes (Table 6). Among all athletes of both sexes who used massage as a method of recovery, the most used technique was hand massage performed by a massage therapist for 30–60 min, at a frequency of 1–2 times a week or more than 4 times a week (Table 7). There was no significant difference between men and women in type of massage, session duration. Men used massage more than 4 times a week significantly more often than women, *p* < 0.001, OR = 4.57, 95% CI 2.07–10.1 (Table 6).

### 3.2. Sauna Bathing

Most of the participants engaged in sauna bathing in 30–60-min sessions 1–2 times a week. There was no significant difference between men and women in frequency and duration of sauna (Table 6 and Table 8).

### 3.3. Daytime Nap

The frequency of use of daytime nap as a method of recovery was associated with the sporting discipline (*p* ≤ 0.001) and athlete tier (*p* = 0.015). A pairwise comparison revealed that the frequency of use of daytime nap was significantly different between the national and international tier groups, which was more common among international athletes (*p* ≤ 0.01, OR = 4.2, 95% CI = 1.5–12.0). A statistically significant difference in the frequency of use of daytime nap was observed between athletes competing in racewalking and middle-distance running (*p* ≤ 0.001, OR = 0.13, 95% CI = 0.04–0.39) and between athletes competing in racewalking and long-distance running (*p* ≤ 0.001, OR = 0.15, 95% CI 0.04–0.49), which was more common among racewalkers (Table 6).

### 3.4. Long Night Sleep

The frequency of use of long night sleep as a method of post-exercise recovery was significantly related to the tier of the athletes (*p* ≤ 0.05). A pairwise comparison revealed that the frequency of use of long night sleep was significantly different between the national and elite tier groups, which was significantly more common among elite-tier athletes (*p* ≤ 0.01, OR = 11.3, 95% CI = 1.3–95.2) (Table 6).

### 3.5. Compression Garments

The frequency of use of compression garments as a method of post-exercise recovery was significantly related to the tier of the athletes (*p* ≤ 0.01). A pairwise comparison revealed that the frequency of use of compression garments was significantly different between the national and international tier groups, which was more common among international athletes (*p* ≤ 0.01, OR = 7.6, 95% CI = 1.6–36.3). The use of compression sportswear was significantly associated with age (*p* ≤ 0.01), which was more common among older athletes according to the results of Mann–Whitney U test (Table 6).

### 3.6. Cold Water Immersion

The frequency of use of cold water immersion as a method of post-exercise recovery was significantly related to the sporting discipline (*p* ≤ 0.001). A pairwise comparison revealed that the frequency of use of cold water immersion was significantly different between athletes competing in racewalking and middle-distance running, which was more common among middle-distance runners (*p* ≤ 0.001, OR = 4.9, 95% CI = 1.8–13.2). The difference between athletes competing in middle- and long-distance running was also significant; the cold water immersion method was used more often by middle-distance runners (*p* ≤ 0.05, OR = 0.16, 95% CI = 0.03–0.8).

### 3.7. Combined Usage of Various Recovery Methods

We examined the possible combinations of the recovery methods (cold immersion, massage, sauna bathing, daytime nap, long night sleep, and compression garments). Overall, 147 athletes used two or more methods of recovery (Table 9). The most common combinations were as follows:(a)massage, sauna bathing, daytime nap, and long night sleep (38.6%, combination 1),(b)massage, sauna bathing, and daytime nap (20.9%, combination 2),(c)massage and sauna bathing (7.8%, combination 3).

The rest of the combinations were used by seven or fewer people, therefore, only three indicated strategies were included in the further analysis. According to the Chi-square test, the use of one of these three strategies was not associated with gender (*p* = 0.21) and according to the Kruskal–Wallace test, the groups did not differ in age (*p* = 0.99). However, the use of one strategy or another was associated with the level of athletes (*p* = 0.029) and discipline (*p* = 0.025). In relation to sport discipline, athletes doing walking are significantly more likely to use strategy 1 (*p* = 0.005) and strategy 2 (*p* = 0.003) than athletes doing middle distances. A similar significance was revealed when comparing athletes doing walking versus staying running in relation to strategy 2 (*p* = 0.016) and strategy 1 (*p* = 0.009). International athletes are more likely to choose strategy 2 (*p* = 0.037) than national athletes. High level athletes more often choose strategy 2 than international *p* = 0.022) and national (*p* = 0.012) athletes.

## 4. Discussion

Our findings demonstrated that sauna bathing, massage, daytime nap, and long night sleep were commonly used by Russian top-tier track and field endurance athletes as post-exercise recovery methods (96.7%, 86.9%, 81%, and 61.4%, respectively). Massage and sauna bathing were frequently complemented by daytime nap and long night sleep. However, recovery methods with proven effectiveness such as cold water immersion and compression garments were rarely used (15% and 7.8%, respectively).

One of the main findings of our study was that the frequencies of use of various recovery methods were increased with the tier of the athletes. For instance, international athletes frequently used sauna bathing, massage, long night sleep, and daytime nap but not cold water immersion and compression garments as recovery methods; however, all these methods except for cold water immersion were widely used among elite-tier athletes. The use of compression garments was increased with the athlete age and tier despite its relative rarity.

The low frequency of use of cold water immersion and compression garments in our study may be attributed to the low awareness of athletes and coaches about their possible effectiveness. Compression garments are widely used by endurance athletes in other countries [25]. In a meta-analysis by Machado et al., diving in cold water for 60 min after the end of the exercise was also confirmed as a means of reducing the severity of DOMS (when compared with passive recovery) and it was concluded that there is a dose–response relationship. The most effective protocol for both rapid and delayed effects, according to the authors, is the use of water at a temperature of 11–15 °C for 11–15 min [23]. Bleakley et al. also showed that cold water immersion can reduce delayed muscle soreness after exercise compared to passive interventions that include rest or no intervention [24].

In addition, several studies have provided evidence for the benefits of daytime nap, which is relevant to both 20-min naps before running and longer 40- and 90-min daytime naps [26,27]. The widespread use of daytime nap in the study group indicated its effectiveness, especially in the twice-a-day exercise regimen often used by endurance athletes while preparing for competitions. However, its effect on recovery rates requires further investigation.

In the studies carried out, one of the most frequently used methods of restoration was massage, for which there is a sufficient amount of information about its effectiveness. Currently, there is insufficient information on the effectiveness of massage as a means of restoring and increasing physical performance. Several studies that reviewed the literature on the effectiveness of pre- and post-exercise massage have been published in recent years. Best et al. reviewed 27 studies (17 case series and 10 randomized controlled trials). The authors found some evidence of the benefits of using massage as a means of recovery from intense exercise [28]. A meta-analysis by Poppendieck et al. [29], which included 22 studies, also provided limited information on the effectiveness of both machine massage and massage by hand as methods of post-exercise recovery. According to their findings, the optimal duration of massage is 5–12 min; the effectiveness of such a regimen is most evident in the immediate recovery period (up to 10 min).

In 2020, a meta-analysis was published by Davis et al., which included 29 studies on pre- and post-exercise massage by hand. The authors also found no evidence supporting the theory that sports massage improves performance; however, it may improve flexibility and reduce DOMS [30]. Massage is still widely used by professional athletes despite the lack of evidence on its effectiveness. According to the available data, more than 70% of French and Spanish top-tier soccer teams use it for recovery [31]. It is also widely used as a recovery method in rugby, track and field athletics, and many other sports. More than 90% of top-tier Russian endurance athletes use massage by hand for recovery. These athletes consider it as one of the most effective recovery methods; the most common regimen is one or two 30–60-min sessions a week.

The discrepancy between the evidence showing the insufficient effectiveness of massage and its widespread use by professional athletes may be attributed to the difference between the protocols used in the studies and those used by the athletes. Thus far, all studies on the effect of massage on recovery only considered massage sessions that lasted for no more than 40 min (often less than that). In most cases, massage was performed once immediately following exercise. Our study showed that athletes mostly preferred massage by hand performed by a massage therapist in 30–60-min sessions, at a frequency of 1–2 times a week or more than 4 times a week. It should be noted that the vast majority of studies evaluated amateur athletes or only physically active people rather than professional athletes; this may also introduce bias to the results.

In contrast to massage, sleep, and cold water immersion, and the use of compression garments, there is currently no conclusive evidence on the effectiveness of sauna use as a post-exercise recovery method. Most studies involving athletes have shown that sauna bathing could effectively increase physical performance and heat acclimation in endurance athletes; however, there is no evidence to support its effectiveness as a method of post-exercise recovery [5,6]. Sauna bathing has various physiological effects. There is ample evidence in support of the beneficial effects of regular sauna bathing on the cardiovascular, immune, and respiratory systems, as well as the lipid profile of individuals with various levels of physical activity [7,8,32]. Positive effects on the cardiovascular system may be associated with vascular dilation and reduction of arterial stiffness and blood pressure [33]. In comparison with wet sauna (low temperature and high humidity), dry sauna (high temperature and low humidity) is believed to be better tolerated by young men and women and induces a more gradual change in blood pressure. Most studies have analyzed the effects of dry sauna bathing [8,34]. However, there is no high-quality research on the effect of sauna bathing on the post-exercise recovery of top-tier endurance athletes.

Another common post-exercise recovery method among Russian track and field athletes was sauna bathing. Usually, it was used 1–2 times a week, and the most frequent durations were 30–60 min and more than 60 min. In a few studies on the effect of sauna bathing on recovery, contradictory results were obtained; however, different protocols were used. Podstawski et al. noted that young, trained men exhibited a decrease in cortisol levels but not testosterone and prolactin levels after four 12-min sauna bathing sessions (90–91 °C, 14–16% relative humidity), interspersed by 1-min sessions of cold water immersion [35]. However, a study by Scoon et al. [36] involving a cohort of long-distance runners showed that 30-min sessions of sauna bathing for 3 weeks could lead to a 32% improvement in running time before exhaustion; this may be equivalent to a 1.9% improvement in race time. The outcome may be associated with a 7.1% increase in plasma volume and a 3.5% increase in red blood cell volume compared with those in the control group. On the other hand, Skorski et al. showed that three 8-min sessions of post-exercise sauna bathing at 80–85 °C had a detrimental effect on the swimmers’ times. The authors concluded that coaches should be mindful when choosing sauna bathing for post-exercise recovery [37].

A study published in 2020 involving 27 young men with prehypertension showed that sauna bathing immediately after endurance exercises was associated with improved blood pressure and increased plasma volume the following day [38]. Another study by the same authors showed that sauna bathing itself may be considered as a strenuous exercise that reduces the maximum load during isometric leg press and bench press and should not be recommended for at least 24 h before the next workout. At the same time, the study showed that sauna bathing immediately after exercise did not affect hormonal changes induced by various types of exercise [39].

Therefore, it can be hypothesized that an immediate positive effect on endurance associated with the increase in the plasma and red blood cell volume, rather than improved post-exercise recovery, may be the reason for the regular use of sauna bathing in the analyzed cohort. In specific situations, e.g., during preparation for events held under high temperature conditions, sauna bathing can also serve as a method of passive heat acclimation.

The limitations of this study include the cross-sectional nature of our work. In addition, some recovery methods, such as the consumption of certain products (proteins, branched chain amino acids, etc.) or the use of foam rollers, were not analyzed. Future studies should assess the effect of sauna bathing and massage, as well as their combination, on post-exercise recovery, speed, strength, and endurance according to the actual protocols used by top-tier athletes under various ambient conditions (different altitudes, humidity levels, temperatures, etc.).

## 5. Conclusions

Sauna bathing, massage, daytime nap, and long night sleep were widely used as methods of post-exercise recovery (96.7%, 86.9%, 81.0%, and 61.4%, respectively). Massage and sauna bathing were the most used methods of post-exercise recovery among Russian endurance track and field athletes. In most cases, they were used in conjunction with daytime nap and long night sleep. Overall, recovery methods were used more often when the tiers of the track and field athletes were higher. International athletes frequently used sauna bathing, massage, long night sleep, and daytime nap but not cold water immersion and compression garments as recovery methods; however, all these methods except for cold water immersion were widely used among elite-tier athletes.

## Figures and Tables

**Figure 1 ijerph-18-11698-f001:**
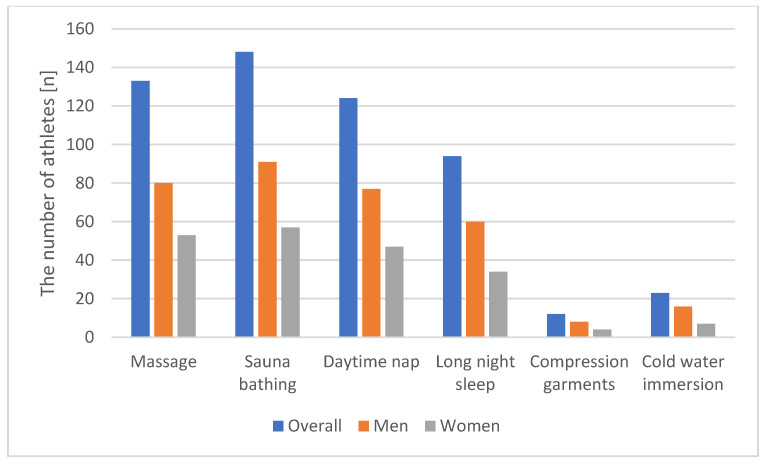
The use of recovery technique according to athletes’ gender.

**Figure 2 ijerph-18-11698-f002:**
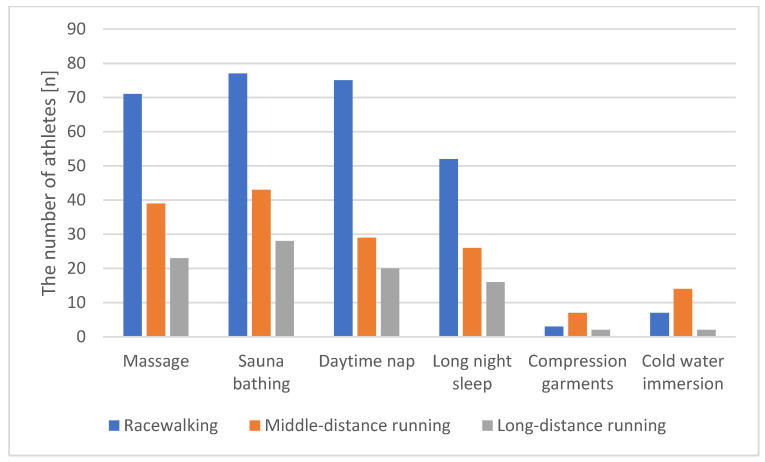
The use of recovery technique according to athletes’ discipline.

**Figure 3 ijerph-18-11698-f003:**
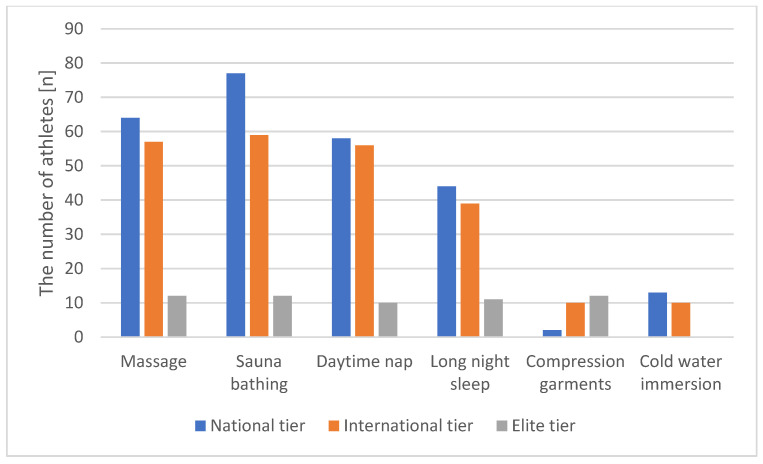
The use of recovery technique according to athletes’ level.

**Table 1 ijerph-18-11698-t001:** The questionnaire “Methods of recovery among highly competitive athletes training. Endurance”.

Gender:	Age
☐ men	Weight (kg)
☐ women	Height (cm)
Main discipline:	Qualification:
☐ medium distances (800–5000 m)	☐ national level
☐ distance run (10,000 m and above)	☐ international level
☐ race walking	☐ winners of the European, World and Olympic Championships
What recovery methods do you regularly use (have used): *(*Select all the options you use)	
☐ massage	☐ active recovery
☐ bath/sauna	☐ cold baths
☐ long night sleep (at least 8 h of continuous sleep)	☐ compression underwear
☐ daytime sleep (30–60 min during the day)	Other________________
Which of the recovery methods used above do you consider the most effective (name three in descending order of effectiveness): __________________________________________________________________________	

**Table 2 ijerph-18-11698-t002:** Age, height, weight, and sex of athletes according to athletes in different sport disciplines.

	Overall (*n* = 153)	Men (*n* = 94)	Women (*n* = 59)
Age (years)	22.7 ± 4.6	22.5 ± 4.9	22.8 ± 4.1
Weight (kg)	57.7 ± 9.0	61.9 ± 8.6	51.0 ± 4.8
Height (cm)	171.9 ± 8.3	175.4 ± 8.0	166.3 ± 5.1
**Sport disciplines**			
Race walking, *n* (%)	80 (52.3)	52 (55.3)	28 (47.5)
Middle-distance, *n* (%)	44 (28.8)	25 (26.6)	19 (32.2)
Long distance, *n* (%)	29 (18.9)	17 (18.1)	12 (20.3)

**Table 3 ijerph-18-11698-t003:** Number of athletes of different tiers.

	National Tier *n* (%)	International Tier *n* (%)	Extra Tier *n* (%)
Men	57 (60.6)	31 (33.0)	6 (6.4)
Women	23 (39.0)	30 (50.8)	6 (10.2)
Racewalking	35 (43.8)	35 (43.8)	10 (12.5)
Middle-distance running	27 (61.4)	16 (36.4)	1 (2.3)
Long-distance running	18 (62.1)	10 (34.5)	1 (3.4)
Overall	80 (52.3)	61 (39.9)	12 (7.8)

**Table 4 ijerph-18-11698-t004:** The questionnaire “The Use of Massage as a Recovery Method Among Highly. Competitive Athletes Exercising Endurance in the Recovery of Athletes”.

How long does the massage usually last? ☐ up to 10 min ☐ up to 30 min ☐ up to 60 min ☐ longer than 60 min
What type of massage have you used the most? ☐ manual massage ☐ self-massage ☐ hardware massage
How often did you use massage when there was such an opportunity at training camps? ☐ 1–2 times a week ☐ 3–4 times a week ☐ more often 4 times a week ☐ less than once a week Other______________
The most commonly used massage was: (Multiple options can be selected) ☐ immediately before the competition (less than 24 h) ☐ on the eve of the competition (24–48 h in advance) ☐ immediately after the hardest workout (up to 12 h after the end) ☐ the next day after the hardest workout (12 or more hours after the end) Other__________________

**Table 5 ijerph-18-11698-t005:** The questionnaire “The use of sauna as a recovery method among highly. competitive athletes training endurance in the recovery of athletes”.

How often did you visit the bath/sauna, if there was such an opportunity at training camps? ☐ 1–2 times a week ☐ 3–4 times a week ☐ more often 4 times a week ☐ less than once a week ☐ did not attend Other________________
What was the most frequent duration of the bath/sauna visit (total time spent in the steam room): ☐ up to 10 min ☐ up to 30 min ☐ up to 60 min ☐ more than 60 min
Most often the bath/sauna was visited (Multiple options can be selected) ☐ immediately before the competition (less than 24 h) ☐ on the eve of the competition (24–48 h in advance) ☐ immediately after the most difficult workout (up to 12 h after its end) ☐ the next day after the hardest workout (more than 12 h after its end) Other______________________

**Table 6 ijerph-18-11698-t006:** Frequency of use of various methods of recovery.

	Massage *n* (%)	Sauna Bathing *n* (%)	Daytime Nap *n* (%)	Long Night Sleep *n* (%)	Compression Garments *n* (%)	Cold Water Immersion *n* (%)
Overall	133 (86.9)	148 (96.7)	124 (81.0)	94 (61,4)	12 (7.8)	23 (15.0)
Men	80 (85.1)	91 (96.8)	77 (81.9)	60 (63.8)	8 (8.5)	16 (17.0)
Women	53 (89.8)	57 (96.6)	47 (79.7)	34 (57.6)	4 (6.8)	7 (11.9)
*p*	0.650	0.950	0.120	0.590	0.150	0.750
Racewalking	71 (88.8)	77 (96.3)	75 (93.8)	52 (65.0)	3 (3.8)	7 (8.8)
Middle-distance running	39 (88.6)	43 (97.7)	29 (65.9)	26 (59.1)	7 (15.9)	14 (31.8)
Long-distance running	23 (79.3)	28 (96.6)	20 (69.0)	16 (55.2)	2 (6.9)	2 (6.9)
*p*	0.40	0.90	<0.001	0.60	N/A	N/A
National tier	64 (80.0)	77 (96.3)	58 (72.5)	44 (55.0)	2 (2.5)	13 (16.3)
International tier	57 (93.4)	59 (96.7)	56 (91.8)	39 (63.9)	10 (16.4)	10 (16.4)
Elite tier	12 (100)	12 (100)	10 (83.3)	11 (91.7)	12 (100)	0 (0)
*p*	0.024	0.790	0.015	0.046	N/A	N/A

N/A—not applicable, chi-square cannot be used because of limited observation amount.

**Table 7 ijerph-18-11698-t007:** Massage: type, length, and frequency.

	Men *n* (%)	Women *n* (%)	*p*	Overall *n* (%)
**Type of massage**				
By hand	75 (93.8)	52 (98.1)	0.24	127 (95.5)
Machine	2 (2.5)	0 (0)	0.25	2 (1.5)
Self-massage	3 (3.8)	1 (1.9)	0.54	4 (3.0)
**Session duration**				
Up to 10 min	2 (2.5)	0 (0)	0.25	2 (1,5)
Up to 30 min	14 (17.5)	8 (15.1)	0.715	22 (16.5)
Up to 60 min	60 (75.0)	40 (75.5)	0.95	100 (75.2)
Longer than 60 min	4 (5.0)	5 (9.4)	0.32	9 (6.8)
**Frequency**				
Less than weekly	6 (7.5)	3 (5.7)	0.68	9 (6.8)
1–2 times a week	22 (27.5)	23 (43.4)	0.06	45 (33.8)
3–4 times a week	11 (13.8)	8 (15.1)	0.83	19 (14.3)
More than 4 times a week	41 (51.2)	17 (32.1)	<0.001	58 (43.6)

**Table 8 ijerph-18-11698-t008:** Frequency and duration of sauna bathing.

	Men *n* (%)	Women *n* (%)	Overall *n* (%)
Frequency			
1–2 times a week	87 (956)	56 (98.2)	143 (96.6)
3–4 times a week	4 (4.4)	1 (1.8)	5 (3.4)
Duration			
Up to 10 min	5 (5.6)	1 (1.8)	6 (4.1)
Up to 30 min	26 (28.9)	17 (29.8)	43 (29.5)
Up to 60 min	39 (43.3)	23 (40.4)	62 (42.2)
Longer than 60 min	20 (22.2)	16 (28.1)	36 (24.5)

**Table 9 ijerph-18-11698-t009:** The number of athletes using different recovery strategies.

Type of Athlete	Strategy 1	Strategy 2	Strategy 3
National tier	17	9	23
International tier	15	2	26
Elite tier	0	1	10
Racewalking	22	2	38
Middle-distance running	6	6	11
Long-distance running	4	4	10

## Data Availability

This statement if the study did not report any data.

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
