# Peer review of "The Prevalence of Use of Various Post-Exercise Recovery Methods after Training among Elite Endurance Athletes"

_ijerph, 2021, doi:10.3390/ijerph182111698_

Round 1

Reviewer 1 Report

Dear

Many parts have been revised. thank you. However, I politely request a little more supplementation.

Table 6. 2 or 3 decimal places, be consistent.

Please provide a table or figure for '3.3 daytime nap'.

Please provide a table or figure for '3.4 Long night sleep'.

Please provide a table or figure for '3.5 Compression garments'.

Please provide a table or figure for '3.6 Cold water immersion'.

Please provide a table or figure for the '3.7' Combined usage of various recovery methods.

Please organize your paragraphs of introduction and discussion. It is recommended that a paragraph consist of 4-5 sentences.

Lines 153 and 155: 81%, insert 15% decimal point (eg 81.0% and 15.0%). Be consistent throughout the document (including abstracts).

There is a mix of p<0.001 and p=0.001 throughout the document. Be consistent.

Reviewer 2 Report

My recommendation is:
Line 35-36 is too general this conclusion, I recommend rewriting focused on concrete results.

Author Response

See attached file.

This manuscript is a resubmission of an earlier submission. The following is a list of the peer review reports and author responses from that submission.

Round 1

Reviewer 1 Report

Thank you for submitting to IJERPH.

This study is a study of athlete recovery, a common recovery method for athletes, but scientific evidence is low. This study is expected to contribute to the field of player recovery from this point of view. However, I hope you will revise more scientific and logic for publication.

Thank you for submitting to IJERPH.

This study is a study of athlete recovery, a common recovery method for athletes, but scientific evidence is low. This study is expected to contribute to the field of player recovery from this point of view. However, I recommend authors will revise more scientific and logic for publication.

Author Response

Response to Reviewer 1

Dear Reviewer. Great thanks for your positive approach and very valuable comments. We tried to improve the text according to your suggestions and we hope that it will satisfy you. All comments were corrected in accordance with the Reviewer's recommendations and marked in yellow in the original document.

Thank you for submitting to IJERPH.

This study is a study of athlete recovery, a common recovery method for athletes, but scientific evidence is low. This study is expected to contribute to the field of player recovery from this point of view. However, I hope you will revise more scientific and logic for publication.

Overall, I would like to ask you to clarify the focus of this study.
I don't understand whether this study is focused on saunas or whether it's examining various recovery in athletes.

Thank you for your expert opinion and we changed and added some text into abstract and end of the introduction paragraph. We changed also the title which now is more precise according to the methods used.

Abstract and Introduction

Physiological analyzes or questionnaires can be used in this study to explain the superiority of saunas and massages.

If the purpose of this study is various recovery methods, please describe it more objectively. This applies throughout the abstract, introduction and discussion.

Thank you for your expert opinion and we changed and added text.

Please, Replace 54.8% to 94 and 45.2% to 59 in your abstract. Put the statistical significance and p-value in the abstract.

Thank you for your expert opinion and we changed and added

Method

I recommend breaking up the paragraphs as follows. 2. Materials and Methods

2.1. Study procedure 2.2.
2.3.
2.5. Data analysis

Lines 107-112 are statistical analysis.

Thank you for your expert opinion and we changed the text and added three subparagraphs:

2.1. Participants

2.2. Inclusion and exclusion citeria

2.3. Questionnaire

2.4.. Statistical analysis

It is recommended to diagram the inclusion and exclusion criteria process for athletes to better understand the study flow.

Thank you for your expert opinion and we special paragraph in “Material and Methods” section.

How was the questionnaire structured? Is it scientifically valid and certified? Or is it the author's arbitrary composition? If the composition is arbitrary, please describe the principles of questionnaire composition, expert advice, and reference to prior literature.

Thank you for your expert opinion. The questionnaires used in the study have not yet been developed in the literature. It is an original work and results from the experience and knowledge of all the authors of this work. After the questionnaire was created, it was consulted with experts from the Department of Sports Medicine at the Sechenov University in Moscow. During the talks and analyzes, its details were refined. In the initial phase, 10 semi-professional athletes were used by completing the questionnaire. This pre-test indicated that they had no problems understanding and answering all the questions. If the reviewer deems it appropriate to include such a detailed description, they will do so in the next step of the review.

We recommend merging Tables 1, 2, and 3 into one and swapping the horizontal and vertical columns. Ex)

Thank you for your expert opinion and we changed table according to your suggestion.

Result

For the scientificity of this study, put the p-value or OR and CI in the results table 4-6.

Thank you for your expert opinion. However, p-value, OR and CI for significant difference between groups is given in the text for each recovery strategy  table will be too overloaded if we put p-value in it. We added p-value in tables 5 and 6. 

No table corresponding to 3.3 to 3.7 described in the results?

Dear Reviewer. We would like to explain that Table 4 corresponds to 3.3-3.6 results. In paragraph 3.7 since the number of combinations was very large, we do not give a general table, but highlight the three most frequent ones.

Discussion

The discussion is too focused on saunas and massages. Please suggest more studies comparing recovery methods.

Thank you for your expert opinion and we added following text:

1.

2.

Unify the terms Male to 'men' and Female to 'women' throughout the text.

Thank you for your expert opinion and we changed the terms in the whole text.

Reviewer 2 Report

My recommendations are the following:
The conclusion from the abstract to be focused on the results of the study is too generalized. Say that elite athletes are prone, then there should be a comparative study between elite and non-elite, I recommend correction.
Lines 91-93 data are not identical to those in the abstract on the percentages between women and men, I recommend correction.
In section 2 I recommend to mention how the selection of these sportswomen was made, which were the exclusion criteria.
I recommend mentioning how many questions the questionnaire contained, the form of the answers, and calculating the Cronbach's α coefficient for the entire questionnaire.
Line 107 recommends the introduction of Analyzed Data sections.
After reviewing your article, no strategy emerges, only the methods used in recovery. In this sense, I recommend a total revision of the article, and the replacement of the word strategy with methods, even in the title. Or a presentation of strategies.
I recommend that in the Introduction and Discussions section several bibliographic sources be mentioned, in order to give weight to this article.

Author Response

Response to Reviewer 2

Dear Reviewer. Great thanks for your positive approach and very valuable comments. We tried to improve the text according to your suggestions and we hope that it will satisfy you. All comments were corrected in accordance with the Reviewer's recommendations and marked in yellow in the original document.

The conclusion from the abstract to be focused on the results of the study is too generalized. Say that elite athletes are prone, then there should be a comparative study between elite and non-elite, I recommend correction.

Thank you for your expert opinion and we changed the text in the abstract.

Lines 91-93 data are not identical to those in the abstract on the percentages between women and men, I recommend correction.

Thank you for your precise opinion and we corrected the mistake. Sorry for that.

In section 2 I recommend to mention how the selection of these sportswomen was made, which were the exclusion criteria.

Thank you for your expert opinion and we added special paragraph.

We have to add precise conditions

I recommend mentioning how many questions the questionnaire contained, the form of the answers, and calculating the Cronbach's α coefficient for the entire questionnaire.

Thank you for interesting comment. There were 14 questions in total, the analysis included 12 questions. The questions concerning the most and least effective methods were not analyzed in this work. The coefficient for 12 questions was 0.31. We explain this by the fact that the questions on the application of the strategy were divided into yes/no, which lowers the coefficient"

Line 107 recommends the introduction of Analyzed Data sections.

We are very sorry but we do not understand fully this comment. Explain please in more detail and we immediately include your comment.
After reviewing your article, no strategy emerges, only the methods used in recovery. In this sense, I recommend a total revision of the article, and the replacement of the word strategy with methods, even in the title. Or a presentation of strategies.

Thank you for your expert opinion and we changed to “methods” in the text.

I recommend that in the Introduction and Discussions section several bibliographic sources be mentioned, in order to give weight to this article.

Thank you for your expert opinion and we added sources in all parts of text.

Round 2

Reviewer 1 Report

I am so sorry. I'm still having a hard time understanding this study.

To write Russian in the results of the abstract, put 'Russian' athletes in the research method section.

Introduction
There is too little logic to the purpose of the study. The introduction should state objective facts. However, this document contains a number of speculative statements.

Move or delete line 57 "There is …popularity" to the end.
Delete line 58 "nevertheless".
Lines 58-60 Add a reference to this sentence.
Line 60: What does "That" mean?
Lines 71-74, I don't understand. What does it mean?
Lines 81-93 should be described in the study method and discussion. It doesn't fit the introduction.

Research methods and results
There is no basis for the questionnaire. What is the basis for filling out the questionnaire, what is the background? Do you have a reference?

Statistical: The author did a chi-square analysis and odds ratio, but not in the table. This part is very important. It is to statistically test the value of this study. However, there is no chi-square test, odds ratio, and p-value in the table.

discussion
See various studies based on the results.
Paragraphs are not organized as a whole in the document. Make 4-6 sentences into one paragraph.

Delete line 333: "In summary" .
Line 334 Unify the decimal point. 81.0%